# In Silico Food-Drug Interaction: A Case Study of Eluxadoline and Fatty Meal

**DOI:** 10.3390/ijms21239127

**Published:** 2020-11-30

**Authors:** Annalisa Maruca, Antonio Lupia, Roberta Rocca, Daniel Keszthelyi, Maura Corsetti, Stefano Alcaro

**Affiliations:** 1Dipartimento di Scienze della Salute, Università “Magna Græcia” di Catanzaro, Campus “S. Venuta”, Viale Europa, 88100 Catanzaro, Italy; maruca@unicz.it (A.M.); alcaro@unicz.it (S.A.); 2Net4Science Academic Spin-Off, Università “Magna Græcia” di Catanzaro, Campus “S. Venuta”, Viale Europa, 88100 Catanzaro, Italy; antonio.lupia@net4science.com; 3Dipartimento di Medicina Sperimentale e Clinica, Università “Magna Græcia” di Catanzaro, Campus “S. Venuta”, Viale Europa, 88100 Catanzaro, Italy; 4Division Gastroenterology-Hepatology, Department of Internal Medicine, NUTRIM School for Nutrition and Translational Research in Metabolism, Maastricht University Medical Center, Universiteitssingel 50, 6229 HX Maastricht, The Netherlands; daniel.keszthelyi@maastrichtuniversity.nl; 5NIHR Nottingham Biomedical Research Centre, Nottingham University Hospitals NHS Trust, Nottingham NG7 2UH, UK; maura.corsetti@nottingham.ac.uk; 6University of Nottingham and Nottingham Digestive Diseases Centre, School of Medicine, University of Nottingham, Nottingham NG7 2RD, UK

**Keywords:** eluxadoline, loperamide, conformational analysis, food-drugs interactions, molecular dynamics

## Abstract

Food-drug interaction is an infrequently considered aspect in clinical practice. Usually, drugs are taken together with meals and what follows may adversely affect pharmacokinetic and pharmacodynamic properties, and hence, the therapeutic effects. In this study, a computational protocol was proposed to explain the different assimilations of two *µ-*receptors agonists, eluxadoline and loperamide, with a peculiar pharmacokinetic profile. Compared to loperamide, eluxadoline is absorbed less after the intake of a fatty meal, and the LogP values do not explain this event. Firstly, keeping in mind the different pH in the intestinal tract, the protonation states of both compounds were calculated. Then, all structures were subjected to a conformational search by using MonteCarlo and Molecular Dynamics methods, with solvation terms mimicking the water and weak polar solvent (octanol). Both computational results showed that eluxadoline has less conformational freedom in octanol, unlike loperamide, which exhibits constant behavior in both solvents. Therefore, we hypothesize that fatty meal causes the “*closure*” of the eluxadoline molecule to prevent the exposure of the polar groups and their interaction with water, necessary for the drug absorption. Based on our results, this work could be a reasonable “case study”, useful for future investigation of the drug pharmacokinetic profile.

## 1. Introduction

Food-drug interactions often cause alterations of the pharmacokinetics or pharmacodynamics of a drug or nutritional element [1]. In particular, food can affect the activity of a drug, i.e., the effects are increased or decreased, or it produces a new effect that neither of them makes on its own. Most of the time this happens due to numerous interactions that the drug establishes with other substances and in different conditions: simultaneous intake of more drugs (drug-drug interaction), dietary supplements (drug-nutrient interaction), or food and beverages (drug-food interaction). In addition, food-food interaction creating matrix effects can also influence drug behavior. In general, these aspects are not considered in clinical practice [2,3]. An example is the case of eluxadoline (Viberzi), a new medication approved by the Food and Drug Administration (FDA) for the treatment of adults with irritable bowel syndrome (IBS) with diarrhea [4]. Eluxadoline wields its effects binding the opioid receptors (ORs) located in the enteric circuitry of the gastrointestinal (GI) tract, which has a pivotal role in the transport of fluids and electrolytes and the control of motility. In particular, the drug is an *μ-* and *κ-*opioid receptor agonist (*μ*OR/*κ*OR), and a *δ-*opioid receptor antagonist (*δ*OR) [5]. IBS is a functional GI disorder, currently considered to be a disorder of the gut-brain interaction, characterized by the presence of recurrent abdominal pain associated with altered stool frequency and form [6]. The prevalence of IBS is about 10% in Western countries, and it represents a high cost to health-care services as these patients use health-care services more than the general population [7,8,9]. There are three phenotypes of IBS, classified according to stool pattern: IBS with diarrhea (IBS-D), IBS with constipation (IBS-C), and IBS with a mixed bowel pattern (IBS-M) [6]. IBS-D patients, about one-third of the IBS population, have a low quality of life, which is further impaired by unpredictable bowel function, urgency, and faecal incontinence [10].

To date, the available pharmacological treatments for IBS have substantial limitations.

Loperamide is a *μ*OR agonist, commonly used as an antidiarrheal medicament and accessible as an over the counter. However, it can lead to constipation and exacerbation of abdominal pain. Alosetron and ramosetron, two potent and selective serotonin 5-HT_3_ receptor antagonists, are useful towards the IBS management. Nevertheless, the former is available only in the USA under a risk management program and the latter only in few countries [11,12,13]. Ondansetron, 5-HT_3_ receptor antagonist is useful to control diarrhea but not pain [14]. However, it is only available for *off-label* use with no data on its long-term use.

Eluxadoline has recently been approved for the treatment of IBS-D, but it has been reported that it might have possible interactions with food [4].

Chemically, eluxadoline (5-({[(2S)-2-amino-3-(4-carbamoyl-2,6-dimethylphenyl)propanoyl] [(1S)-1-(4-phenyl-1H-imidazol-2-yl)ethyl]amino}methyl)-2-methoxybenzoic acid) is a phenyl imidazole that employs a unique mechanism for IBS treatment as it works simultaneously binding to human ORs, as a μOR agonist with a binding affinity (Ki) of 1.8 nmol/L, and as a δOR antagonist with a Ki of 430 nmol/L, respectively. At present, the Ki is unknown for the κOR human receptor [4,15,16]. In animal models, eluxadoline has been demonstrated to reduce gut motility and secretion. In particular, in vivo animal models, it reduces gut transit but without completely inhibiting it. [10]. In clinical studies, eluxadoline has been reported to be effective and safe in treating both altered bowel habit and pain in IBS-D patients [10]. Reported adverse events were mainly constipation, nausea, and abdominal pain, [17]. There were some cases of the sphincter of Oddi spasm and pancreatitis. However, these seemed to mainly occur in patients with a history of previous cholecystectomy alcohol abuse [10].

Eluxadoline appears to be a locally active compound with low oral bioavailability. Following oral administration of eluxadoline 100 mg tablets, extremely low levels are found in plasma, and the *C_max_* is decreased by 50% and AUC by 60% when the drug was co-administered with a high-fat meal. Plasma protein binding of eluxadoline was 81% [4]. Eluxadoline is mostly (82%) excreted in feces, and <1% is found in the urine [4]. 

Nowadays, computational techniques represent powerful tools in the pharmaceutical industry and academia [18,19]. Generally, they are widely applied to drug discovery in different applications fields, but they are useful also for the deep comprehension of the ligand selectivity and the targets folding [20,21,22,23,24,25,26,27,28,29,30,31,32]. However, there is a considerable need to be able to predict the clinical behavior of drugs in order to optimize pharmacological treatments by increasing their efficacy and decrease the frequency and/or severity of adverse events.

In this study, we performed the Monte Carlo (MC) conformational search and Molecular Dynamics (MD) simulations, intending to explain the different pharmacokinetic profiles of the eluxadoline and the loperamide. Loperamide has been selected as it is the most used OTC drug used in patients presenting with diarrhea worldwide. In addition, loperamide is characterized by similar pharmacokinetic properties, such as high plasma binding (95%) and elimination via feces.

As far as the food-drug interaction of loperamide and eluxadoline is concerned, we were particularly interested in studying the relation with fat and how this can impact drug absorption. In the light of their chemical structures and logP values, it can be assumed that there is no decrease in absorption of eluxadoline after a fatty meal, given its more polar structure compared to loperamide. It is generally well-established that the structure and function of a compound are pH-dependent because the ionizable groups of the drugs change their titration behavior at different pH values. Thus, the most abundant protonation states of these two drugs were calculated, considering the different pH in the intestinal tract. Electrostatic properties ruled different drug properties, such as ligand binding, transport, and absorption but also structure and resistance to enzymatic and chemical degradation [33,34,35]. Moreover, octanol and water were used to simulate the intestinal environment with or without a fatty meal.

## 2. Results

### 2.1. Analysis of Electrostatic Properties

The bioavailability of a compound is influenced by several factors, including logP and the protonation state. Firstly, we compared the experimental logP values of eluxadoline and loperamide, intending to find a correlation with the different absorption after a fatty meal. Unfortunately, the Drug Bank reports the experimental data only for loperamide (logP equal to 5.5) [36]. Therefore, we used a ChemAxon tool to predict the data for both drugs, highlighting a paradox [37]. Eluxadoline showed the lowest value (logP equal to 1.8), by indicating lower lipophilicity than loperamide (logP equal to 4.77). Then, we calculated the most probable protonation states. Indeed, the structure and function of compounds are pH-dependent, due to the changes in the titration behavior of the ionizable group of the drugs at different pH values. For this reason, we considered a value of pH, ranging from 6 to 8, by taking into account the different environment in the intestinal tract. Loperamide showed only an ionization state, which gave a probability of existence equal to 97% in the evaluated pH range (Figure 1a). Conversely, for eluxadoline, we found three different ionization states with a probability of existence between 37 and 85% in the pH range under consideration (Figure 1b–d). Specifically, at pH 6, eluxadoline (E1) exhibited two positive charges at the level of the primary amine and the imidazole ring (Figure 1b). At pH 7, eluxadoline (E2) appeared in the zwitterionic form, characterized by a positive and a negative charge on the primary amine and the carboxyl group, respectively (Figure 1c). Finally, at pH 8 the eluxadoline (E3) was negatively charged due to the deprotonated carboxyl group (Figure 1d).

### 2.2. Monte Carlo (MC)

As described in the Materials and Methods section, two MC simulations were performed with solvation terms mimicking the water and weak polar solvent (octanol). The global minima will be mentioned taking into account the name of the compound (loperamide, **L**, or eluxadoline, **E1**, **E2** and **E3**), the conformational technique (Monte Carlo, **MC**, or Molecular Dynamics, **MD**), the energy rank (**1**,**2**, etc.) and the solvent (water, **w**, or octanol, **o**).

#### 2.2.1. Loperamide

Results from MC analysis of loperamide in water showed 90 unique conformations, which, following minimization fall in 16 energy minima. Only 4 minima obtained a Boltzmann probability (%B) greater than 10% (Table 1 and Appendix A). Conversely, for the loperamide in octanol, we found 52 unique conformations that converged in 15 energy minima. Among these, just 3 minima exhibited a %B > 10% (Table 1 and Appendix A).

As reported in Table 1, the loperamide (**L**) global energy minimum, obtained in water (**L-MC1w**), showed a greater %B (47.24%) compared to other minima (**L-MC2w**, **L-MC3w**, and **L-MC4w**) and can represent almost half of the total population of conformations. **L-MC1w** is characterized by a π-cation interaction between benzene of the biphenyl portion and the quaternary ammonium (Appendix A). In order to examine the differences among the global minimum and the other minima in water, the Root Mean Square deviation (RMSd) was computed on the heavy atoms. This analysis revealed RMSd values higher than 2 Å, thus highlighting the presence of a heterogeneous population of conformations. Among them, only **L-MC2w** exhibited an intramolecular hydrogen bond (H-bond) between the quaternary ammonium, and the carbonyl group (Appendix A). **L-MC3w** and **L-MC4w** have no favorable intramolecular interactions, and they differ from each other only in the 4-hydroxypiperidinil conformation (Appendix A).

On the other hand, the energy minima found in octanol showed a similar probability of existence with %B values ranging between 37.14 and 22.27 (Table 1). Moreover, they showed RMSd values compared to the global minimum higher than 2 Å, by indicating a high structural and geometric heterogeneity of the loperamide in octanol as well. **L-MC1o** and **L-MC2o** are characterized by the same π-cation interaction observed in **L-MC1w** between benzene of the biphenyl portion and the quaternary ammonium, but with a different orientation of the biphenyl moiety (Appendix A). Similarly to **L-MC2w**, the **L-MC3o** exhibited an H-bond between the quaternary ammonium and the carbonyl group (Appendix A).

The geometric comparison among the best loperamide conformations, obtained both in water and octanol, has pointed out that **L-MC2o** and **L-MC3o** are almost similar to **L-MC1w** and **L-MC2w**, respectively (Appendix A). 

#### 2.2.2. Eluxadoline

For the analysis of eluxadoline (**E**), three different ionization states were considered (E1, E2, and E3). 

For E1, 357 unique conformations were generated in water. Following minimization, the similar conformations were deduplicated, giving to 102 energy minima, but only **E1-MC1w** and **E1-MC2w** showed a value of %B > 10% (Table 2 and Appendix A). In octanol, 204 unique conformations were found by leading after minimization to 43 energy minima, whose only 2 obtained a value of %B > 10% (Table 2 and Appendix A). **E1-MC1w** and **E1-MC2w** have a similar probability of existence and a remarkably high structural similarity, as demonstrated by the RMSd value of 0.51 Å (Table 2). They are characterized by an H-bond and a salt bridge between the carboxylic group and the positively charged imidazolium. The remaining polar moieties of both conformations are exposed to the solvent (Appendix A). In octanol, the global minimum (**E1-MC1o**) is characterized by a probability of existence equal to 51.61% and can be considered the most representative. Both **E1-MC1o** and **E1-MC2o** are characterized by several intramolecular interactions with the aim to reduce the contact among polar groups and the weak polar solvent. However, we observed a high structural diversity between **E1-MC1o** and **E1-MC2o**, given the RMSd value equal to 3.71 Å. By analyzing the intramolecular contacts, we observed π-cation and π-π stacking interactions between the 2,6-dimethylphenyl and the imidazole rings for **E1-MC1o**. Moreover, the quaternary ammonium and the carboxyl group of the 2-methoxy-benzoic acid moiety were involved in an H-bond and a salt bridge (Appendix A). In the **E1-MC2o**, the 2-methoxy-benzoic acid moiety and imidazolium ring interact among them with an H-bond and a salt bridge, and with the quaternary ammonium by two π-cation interactions (Appendix A). The structural comparison among the conformations obtained in water and octanol from the MC analysis exhibited RMSd values greater than 2 Å for all structures, highlighting the high structural difference of E1 (Appendix A).

The MC simulation of E2 in water provided 427 unique conformations. The following minimization allowed to deduplicate the identical conformations, leading to 139 energy minima, but only 3 conformations showed a value of %B > 10% (Table 3 and Appendix A). Instead, in the MC simulation performed in octanol, 309 unique conformations were found and, after the multiminimization, 79 energy minima were obtained, whose only 2 structures showed a value of %B > 10% (Table 3 and Appendix A). At pH 7, eluxadoline is characterized by several energy minima with similar %B in both solvents and lower RMSd value respect to the global minima, except for **E2-MC2w** (Table 3 and Appendix A). **E2-MC1w**, **E2-MC1o,** and **E2-MC2o** exhibited the same salt bridge and H-bond between the carboxyl group of the 2-methoxy-benzoic acid and the quaternary ammonium (Appendix A). In the **E2-MC2w**, the carboxyl group and the *p-*amide of the 2,6-dimethylphenyl moiety are involved in the establishment of an H-bond (Appendix A). Conversely, **E2-MC3w** exposed all polar groups to the solvent without intramolecular interactions (Appendix A). 

For E3, 312 unique conformations were generated from MC simulation in water. Following minimization, 84 energy minima were obtained, but only 3 conformations showed a %B value > 10% (Appendix A). In octanol, the MC simulation generated 204 unique conformations, that converged in 43 energy minima, whose 2 with a %B value > 10% (Table 4 and Appendix A). In water, we observed low %B values, also for the global energy minimum, and high RMSd, by pointing out high conformational freedom. Conversely, in octanol, the two energy minima showed %B more significant than 40% and an RMSd value between them of 0.61 Å. As previously observed for E1, E3 showed high RMSd values among the energy minima both solvents, by highlighting the engagement of completely different conformations (Appendix A). 

For the simulations results in water, **E3-MC1w** is characterized by one π-π interaction and two H-bonds, while no intramolecular interactions were observed for **E3-MC2w** and **E3-MC3w** (Appendix A). Specifically, the two substituted aromatic function of the E3 are involved in one π-π interaction and H-bonds among them, while another H-bond is established by carbamoyl moiety with the imidazole ring (Appendix A). In octanol, **E3-MC1o** and **E3-MC2o** are characterized by a strong “closed” conformation in which the polar groups are inclined to hide from the solvent, by generating the most significant number of intramolecular interactions between them (Appendix A).

From MC analysis, we can conclude that eluxadoline, at pH 6 and 8, with a charged structure, is inclined to agglomerate in a weak polar solvent to prevent the exposure of charged groups.

### 2.3. Molecular Dynamics (MD)

As described in the Materials and Methods section, two systems were submitted to MD simulations with solvation terms mimicking the water and weak polar solvent (octanol). These two co-ordinate investigations, firstly, held only one molecule (*scenario 1*), and secondly, two molecules (*scenario 2*). The aim was to understand and possibly observe a different behavior between loperamide and eluxadoline. 

#### 2.3.1. Loperamide

***Scenario 1.*** The analysis of the ligand torsions allowed us to summarize the conformational evolution of every rotatable bond (RB) throughout the simulation trajectories. The loperamide torsion profile exhibited similar behavior in both solvents, except for the *α* and *β* torsions, highlighted in red and lilac, respectively (Appendix A). The *α* RB between the quaternary ammonium to the biphenyl moiety has a better torsional profile in water with respect to octanol, contrarily to *β* RB (amide torsion), which assumes more further angle values in octanol than in water. These observations suggest that loperamide embraces similar conformational space in both solvents. 

Moreover, the conformational changes of loperamide, during the MD simulation, were evaluated and computed into the RMSd matrix (Appendix A). From this analysis came out the very shallow conformational variety of loperamide, with a mostly extensive conformation in both solvents. This allows the constant exposure of polar functional groups to the solvent. All MD frames were submitted to minimization, in order to deduplicate similar conformations. Thus, 3 and 2 energy minima with a %B > 10 were identified in water and in octanol, respectively (Table 5). No intramolecular interactions were observed, unlike the global minima obtained with the MC (Figure 2).

***Scenario 2.*** By examining the MD simulation of the solvated boxes containing two molecules of loperamide, not one intermolecular interaction between the two compounds were observed in both water and octanol. As shown in Appendix A, only an occasional H-bond was observed during all MD trajectory in water, because loperamide prefers to interact with the solvent. Indeed, we observed that loperamide established several H-bonds and van der Waals interactions with water and octanol, respectively.

#### 2.3.2. Eluxadoline

***Scenario 1.*** At pH 6, the torsional profile of E1 (Appendix A) pointed out a different behavior of some RB throughout the simulation trajectory in the two solvents. Most of the E1 RB showed less degree of freedom in octanol than in water because the torsions exhibiting polar groups are involved in intramolecular interactions. These events lead to prevent unfavorable interactions with the hydrophobic solvent and consequently, the possibility of rotations. In particular, the *γ* and *ζ* torsions, encoded in purple and light green, respectively, showed a greater degree of freedom in water than in octanol. On the other hand, the ε and *δ* torsions, encoded in light green and light orange, respectively, explored different conformations in both solvents (Appendix A). 

At pH 7 and pH 8 we also observed RB exhibiting different profiles (Appendix A). Specifically, for E2 *δ* and *γ* torsions, colored in light orange and purple, respectively, visited a different conformational space in water and octanol (Appendix A). For E3, a significant difference was observed for *γ* and *ζ* torsions, encoded in brown and pink, respectively, since they exhibited a better torsional profile in water than in octanol (Appendix A). 

In order to achieve a collection of information on the degree of E1 flexibility, the conformational space was explored by MD simulation, and the resulting trajectories were used to build the RMSd matrices. The graphical analysis of matrices highlighted a more significant heterogeneous population of conformations in water than in octanol. Specifically, three main structures can be counted throughout the dynamics in octanol, with a final conformation which remains stable in the last part of the MD (Appendix A). Similarly, at pH 8, E3 revealed a greater structural variety in water than in octanol (Appendix A) and the maintenance of the same structure for all MD simulation. On the contrary, at pH 7, eluxadoline behaved very similarly in MD simulations conducted in octanol and water. In fact, the RMSd matrices pointed out that E2 visited two more important conformations both in water and in octanol (Appendix A). 

As for loperamide, all the conformations collected by the MD were subjected to minimization to deduplicate the similar structures (Table 6). By confirming RMSd matrix analysis, a greater structural variety was observed for E1 in water, where three energy minima with a similar %B value were collected. Conversely, in octanol, we found only two energy minima, of which the global one has a % B equal to 82.86%, by representing the most frequent structure in MD simulation. 

Interestingly, all minima showed an intramolecular H-bond between the carboxyl group of the 2-methoxy-benzoic acid and the *p-*amide of the 2,6-dimethylphenyl moiety, while an additional π-cation was observed in **E1-MD1w** and **E1-MD3w** between the positively charged imidazolium and the aromatic function of 2,6-dimethylphenyl ring (Figure 3a–c).

Regarding the analysis of E2, we found three energy minima with a %B value > 10 in water. In particular, the global minimum (**E2-MD1w**) is characterized by an intramolecular H-bond between the carboxyl group of the 2-methoxy-benzoic acid and *p-*amide of the 2,6-dimethylphenyl moiety, as observed for **E1-MD1w** (Figure 4a). No intramolecular interactions were noted in **E2-MD2w** and **E2-MD3w**, all of which showed RMSd values of more than 0.80 Å with respect to the global minimum (Figure 4b,c and Table 7). In octanol, 2 energy minima with RMSd equal to 0.51 Å were found, suggesting a high structural similarity. Moreover, **E2-MD1o** and **E2-MD2o** are characterized by a closer conformation, exhibiting an H-bond and electrostatic interaction between the carboxyl group of the 2-methoxy-benzoic acid and the quaternary ammonium (Figure 4d,e).

Finally, four energy minima and a unique global minimum with a %B of 96.7% were found for E3 in water in octanol, respectively (Table 8). Therefore, the protonation state of eluxadoline at pH 8 showed a large conformational variety in the water with all the polar groups exposed to the solvent with the aim to establish the largest intermolecular interactions. Only the global minimum (**E3-MD1w**) exhibited a π-π interaction between the aromatic portion of the imidazole ring and the 2,6-dimethylphenyl moiety, while **E3-MD2w**, **E3-MD3w** and **E3-MD4w** are characterized by an open conformation without intermolecular interactions (Figure 5a–d). Conversely, octanol E3 tends to close its structure, moving the polar groups away from the solvent. Specifically, the *p-*amide of the 2,6-dimethylphenyl moiety interacted with the carboxyl group of the 2-methoxy-benzoic acid to establish an H-bond (Figure 5e).

***Scenario 2.*** In contrast to scenario 2 of loperamide, all the three protonation states of eluxadoline exhibited a greater number of intermolecular interactions (i.e., H-bonds) between the two molecules during the MD simulation performed in water (Appendix A). 

For E1, the most important intermolecular interactions involved the charged groups, i.e., the imidazolium ring, the carboxyl groups, and the ammonium. Both E1 structures are arranged in the space allowing for the two carboxyl groups involved in H-bonds and salt bridges with the quaternary ammonium and the imidazolium ring (Figure 6a,b). Two molecules of E2 established H-bonds between the carboxyl groups and the quaternary ammonium and a π-π interaction between the two benzoate rings (Figure 6c,d). Concerning E3, we observed fewer interactions between the two molecules: An H-bond among the imidazole ring and the *p-*amide of the 2,6-dimethylphenyl moiety and a π-π interaction between the 2-methoxy-benzoic acid and the 2,6-dimethylphenyl moiety (Figure 6e,f).

Finally, the “close” conformation taken in octanol from eluxadoline prevents the formation of intermolecular interactions between the two molecules of the drug. Thanks to the hydrophobic features of the eluxadoline, we observed weak van der Waals interaction with the solvent and several intramolecular contacts employing its polar groups.

## 3. Discussion and Conclusions

In this study, a computational protocol was proposed for explaining the different assimilation of two µ-receptors agonists, loperamide and eluxadoline, after a fatty meal. First, the analysis of the protonation state of both ligands was performed, taking into account the pH range along the entire gastrointestinal tract. Loperamide is characterized by only one principal state of protonation, while eluxadoline is conditioned by three protonation states with similar probability of existence. Both computational techniques showed the constant behavior of loperamide in both solvents due to an extended conformation that promotes a suitable network of interactions with water and allows the movement of the molecule from octanol to water. On the other hand, eluxadoline is affected by the change of pH along the entire gastrointestinal tract, because three different protonation states are computed in the pH ranging from 6 to 8. Typically, eluxadoline showed greater conformational freedom in water than octanol, where, instead, it assumes a “closed” conformation that prevents the unfavorable interactions of polar groups with the non-polar solvent. Likewise, to loperamide, eluxadoline exhibited an extended conformation in water, where the charged groups can be well solvated. In light of the observed, the “close” conformation of eluxadoline in octanol hides all the polar groups from the interactions with the solvents, and this event prevents the movement into water. Therefore, although eluxadoline has a lower logP than loperamide, providing for better absorption of the same with a fatty meal, in reality, its conformation prevents the exposure of the polar groups at the interface of the two solvents, trapping eluxadoline in the fat, which means that this can impact the degree to which the drug can interact with the epithelium of the intestine.

In conclusion, our model-based analysis suggests that the fatty meal causes the closure of eluxadoline molecule intending to prevent the exposure of the polar groups at the interphase of both solvents and thus their interactions with water, necessary for the drug absorption. Conversely, loperamide is influenced neither by the presence of octanol or water, or by the variation of the pH. It maintains a constant conformation along the gastrointestinal tract, exposing all the polar groups both in water and in octanol, allowing an easier transition to the interface of the fatty meal and facilitating its absorption. These results offer interesting consideration as to whether this different behavior of the two drugs could explain the lack of effect of eluxadoline in some patients. This case study also suggests that a closer collaboration between clinicians and medicinal chemistry field could benefit clinical practice. The clinicians could discover that discussing issues with drugs with medicinal chemistry experts could help to explore the potential mechanism behind these.

## 4. Materials and Methods

### 4.1. LigPrep and MarvinSketch

The LigPrep program, implemented in Maestro ver. 11.5, was used for building and optimizing the chemical structure of loperamide and eluxadoline [38]. MarvinSketch, a ChemAxon tool, was also employed to calculate logP value and the most predominant protonation state for our ligands, taking into account the change of pH along the intestinal tract, i.e., pH 6, 7 and 8 [37]. Then, all protonation forms of the compounds were energy minimized using OPLS2005 as the force field [39].

### 4.2. Monte Carlo (MC)

The conformational space of both ligands was deeper investigated by the MC stochastic methodology, using MacroModel ver. 11.9 [40]. This method explores conformational space by starting from a known structure and randomly modifying the rotatable bonds to generate new conformations in an unpredictable order. MC simulations were performed using two solvents, water, and octanol. GB/SA water was selected as an implicit model of solvation for mimicking the physiological condition of the fasting intestinal lumen or the interstitial environments, while octanol simulates the hydrophobic medium induced by the fatty meal. Considering the different protonation states, all rotatable bonds were randomized carrying out 10,000 iterations, using OPLS2005 as force field, since it is widely used for organic molecules [39,41]. All the generated structures were energetically minimized with the Polak-Ribière Conjugate Gradient (PRCG) algorithm, using 5000 steps. All the conformers were structurally compared through their heavy atoms without considering the energy differences but using the RMSd as a similarity criterion and setting a value of 0.5 Å as a *cut-off* to eliminate redundant structures. The resulting unique conformers were clustered and submitted to a Boltzmann probability analysis (%B).

Finally, their geometric analysis was performed on the conformations with a value of %B > 10%.

### 4.3. Molecular Dynamic (MD)

To study the dynamic conformational change over time of loperamide and eluxadoline, MD simulations were carried out with the Desmond software [42]. Moreover, the MD simulations were useful to evaluate the interactions of each ligand with the surrounding environment and with themselves. Two systems were built: 

*Scenario 1*, a single molecule of eluxadoline or loperamide was solvated into water or octanol box; 

*Scenario 2*, two molecules of eluxadoline or loperamide were solvated into water or octanol box.

All systems were solvated in a cubic-shaped box with periodic boundary conditions using the TIP3P water model, and Cl^−^ and/or Na⁺ counterions were added to neutralize the net charge [39,43]. For each system, an MDs of 500 ns was performed under NPT conditions at 1 atm and 300 K, using OPLS2005 as force field. MDs were performed under the following conditions: 500 ns of simulation time under NPT conditions at 1 atm and 300 K; recording interval equal to 500 ps; OPLS2005 as force field. The time step was set to 2 fs. A recording interval of 500 ps both for energy and trajectory was set, obtaining 1000 snapshot saved structures. Then, two types of analyses were carried out, taking into account the energy profile and the geometrical behavior of both ligands. All sampled conformations were energetically minimized with the same protocol previously described in the MC paragraph. 

Finally, the analysis of ligands torsion was performed to evaluate the conformational evolution of each RB throughout the simulation trajectory.

## Figures and Tables

**Figure 1 ijms-21-09127-f001:**
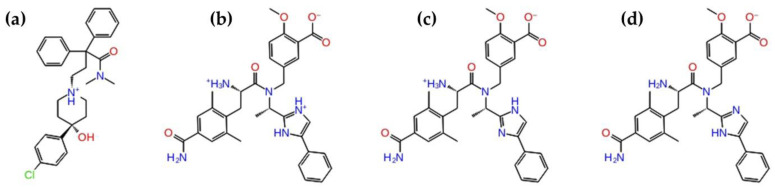
2D structures of (**a**) loperamide and (**b**–**d**) eluxadoline. For eluxadoline, the three most populated protonation state simulated at a range of pH (6–8) corresponding to the gastrointestinal tract were reported: (**b**) eluxadoline at pH 6 (E1), (**c**) eluxadoline at pH 7 (E2) and (**d**) eluxadoline at pH 8 (E3).

**Figure 2 ijms-21-09127-f002:**
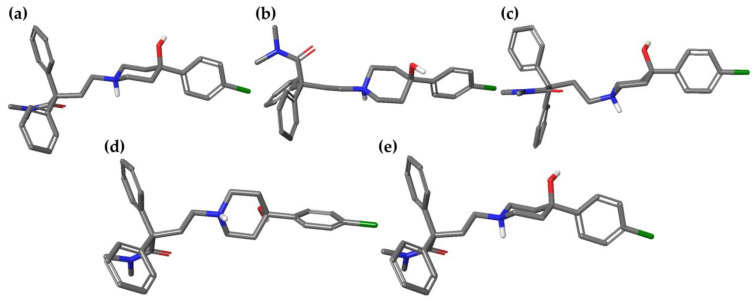
Three-dimensional structures of the best energy minima (%B > 10%) of loperamide (L), obtained after MD simulations conducted in both solvents. (**a**) **L-MD1w**, (**b**) **L-MD2w**, (**c**) **L-MD3w**, (**d**) **L-MD1o** and (**e**) **L-MD2o**.

**Figure 3 ijms-21-09127-f003:**
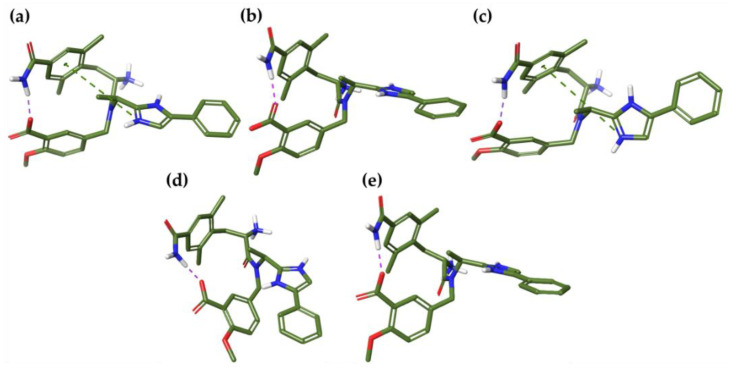
Three-dimensional structures of the best energy minima (%B> 10%) of E1, obtained after MD simulations conducted in both solvents. Intramolecular interactions are shown as green (π-cation) and purple (H-bond) dash lines. (**a**) **E1-MD1w**, (**b**) **E1-MD2w**, (**c**) **E1-MD3w**, (**d**) **E1-MD1o**, and (**e**) **E1-MD2o.**

**Figure 4 ijms-21-09127-f004:**
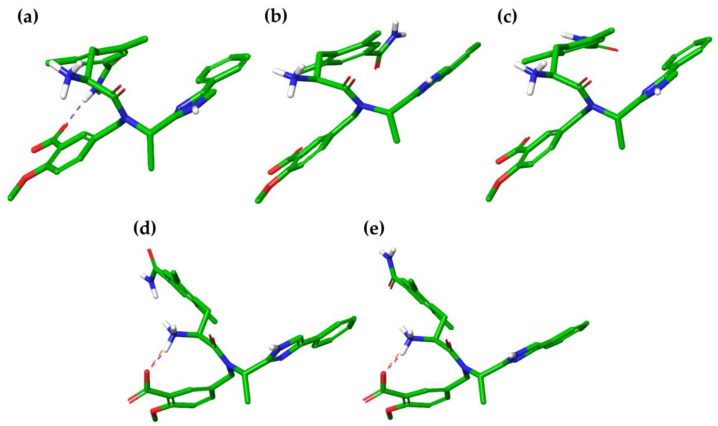
Three-dimensional structures of the best energy minima (%B> 10%) of E2, obtained after MD simulations conducted in both solvents. Intramolecular interactions are shown as green (π-cation) and purple (H-bond) dash lines. (**a**) **E2-MD1w**, (**b**) **E2-MD2w**, (**c**) **E2-MD3w**, (**d**) **E2-MD1o**, and (**e**) **E2-MD2o.**

**Figure 5 ijms-21-09127-f005:**
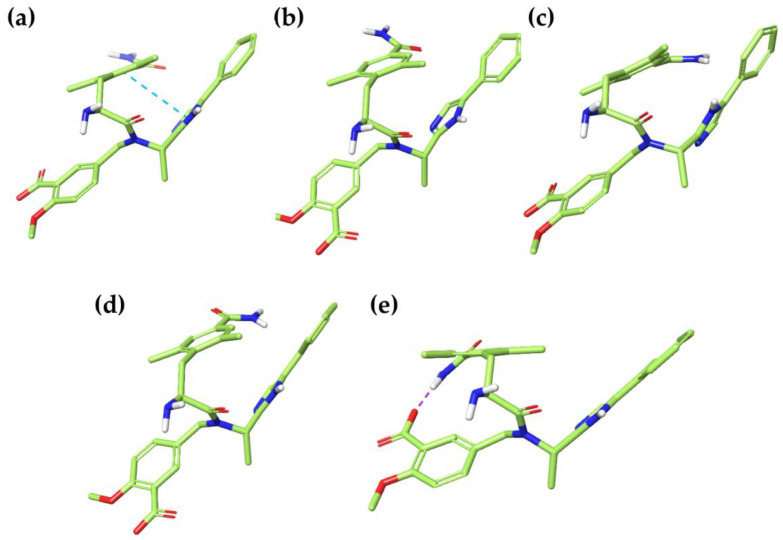
Three-dimensional structures of the best energy minima (%B> 10%) of E3, obtained after MD simulations conducted in both solvents. Intramolecular interactions are shown as cyan (π-π) and purple (H-bond) dash lines. (**a**) **E3-MD1w**, (**b**) **E3-MD2w**, (**c**) **E3-MD3w**, (**d**) **E3-MD4w**, and (**e**) **E1-MD1o**.

**Figure 6 ijms-21-09127-f006:**
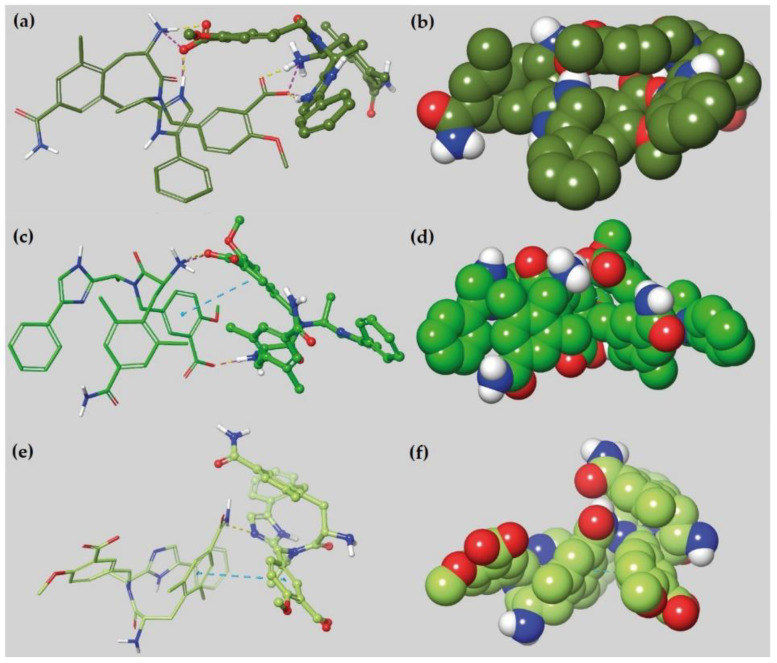
3D representation of two molecules of E1 (**a**,**b**), E2 (**c**,**d**) and E3 (**e**,**f**) interacting during the MD simulations in water. H-bonds, salt bridges and π-π interactions were shown as yellow, magenta, and cyan dash lines, respectively.

**Table 1 ijms-21-09127-t001:** %B and Root Mean Square deviation (RMSd) values of the most probable energy minima for loperamide, obtained after MC simulations conducted both solvents. RMSd values (Å) are calculated on the heavy atoms of loperamide compared to the global energy minima (**L-MC1w** and **L-MC1o**).

Solvent	Minimum Energy	%B	RMSd (Å)
Water	**L-MC1w**	47.24	-
	**L-MC2w**	19.02	2.58
	**L-MC3w**	12.27	2.51
	**L-MC4w**	10.02	2.18
Octanol	**L-MC1o**	37.14	-
	**L-MC2o**	34.31	2.54
	**L-MC3o**	22.27	2.67

**Table 2 ijms-21-09127-t002:** %B and RMSd values of the most probable energy minima for E1, obtained after MC simulations conducted both solvents. RMSd values (Å) are calculated on the heavy atoms of E1 compared to the global energy minima (**E1-MC1w** and **E1-MC1o**).

Solvent	Minimum Energy	%B	RMSd (Å)
Water	**E1-MC1w**	37.31	-
	**E1-MC2w**	31.29	0.51
Octanol	**E1-MC1o**	51.61	-
	**E1-MC2o**	12.05	3.71

**Table 3 ijms-21-09127-t003:** %B and RMSd values of the most probable energy minima for E2, obtained after MC simulations conducted both solvents. RMSd values (Å) are calculated on the heavy atoms of E2 compared to the global energy minima (**E2-MC1w** and **E2-MC1o**).

Solvent	Minimum Energy	%B	RMSd (Å)
Water	**E2-MC1w**	15.73	-
	**E2-MC2w**	13.47	3.53
	**E2-MC3w**	10.66	0.69
Octanol	**E2-MC1o**	18.73	-
	**E2-MC2o**	13.05	0.57

**Table 4 ijms-21-09127-t004:** %B and RMSd values of the most probable energy minima for E3, obtained after MC simulations conducted in both solvents. RMSd values (Å) are calculated on the heavy atoms of E3 compared to the global energy minima (**E3-MC1w** and **E3-MC1o**).

Solvent	Minimum Energy	%B	RMSd (Å)
Water	**E3-MC1w**	18.43	-
	**E3-MC2w**	12.54	3.52
	**E3-MC3w**	12.07	3.79
Octanol	**E3-MC1o**	48.25	-
	**E3-MC2o**	41.22	0.61

**Table 5 ijms-21-09127-t005:** *%*B, RMSd values and intramolecular interactions of the most probable energy minima for loperamide, obtained after the minimization of all MD frames in both solvents. RMSd values are reported in Å, and they are calculated on the heavy atoms of loperamide compared to the global energy minima (**L-MD1w** and **L-MD1o**).

Solvent	Minimum Energy	%B	RMSd (Å)	Intramolecular Interactions
Water	**L-MD1w**	39.24	-	No
	**L-MD2w**	32.59	0.98	No
	**L-MD3w**	28.16	0.93	No
Octanol	**L-MD1o**	58.99	-	No
	**L-MD2o**	40.90	0.93	No

**Table 6 ijms-21-09127-t006:** *%*B, RMSd values and intramolecular interactions of the most probable energy minima for E1, obtained after the minimization of all MD frames in both solvents. RMSd values are reported in Å, and they are calculated on the heavy atoms of E1 compared to the global energy minima (**E1-MD1w** and **E1-MD1o**).

Solvent	Minimum Energy	%B	RMSd (Å)	Intramolecular Interactions ^1^
Water	**E1-MD1w**	33.11	-	1HB, 1EI
	**E1-MD2w**	31.97	1.45	1HB
	**E1-MD3w**	12.67	1.04	1HB, 1EI
Octanol	**E1-MD1o**	82.86	-	1HB
	**E1-MD2o**	13.66	1.88	1HB

^1^ HB: hydrogen bond, EI: electrostatic interactions (π-cation).

**Table 7 ijms-21-09127-t007:** %B, RMSd values and intramolecular interactions of the most probable energy minima for E2, obtained after the minimization of all MD frames in both solvents. RMSd values are reported in Å, and they are calculated on the heavy atoms of E2 compared to the global energy minima (**E2-MD1w** and **E2-MD1o**).

Solvent	Minimum Energy	%B	RMSd (Å)	Intramolecular Interactions ^1^
Water	**E2-MD1w**	41.46	-	1HB
	**E2-MD2w**	38.00	1.14	No
	**E2-MD3w**	15.51	0.81	No
Octanol	**E2-MD1o**	50.15	-	1HB/1EI
	**E2-MD2o**	39.12	0.51	1HB/1EI

^1^ HB: hydrogen bond; EI: electrostatic interactions (π-cation).

**Table 8 ijms-21-09127-t008:** %B, RMSd values and intramolecular interactions of the most probable energy minima for E3, obtained after the minimization of all MD frames in both solvents. RMSd values are reported in Å, and they are calculated on the heavy atoms of E3 compared to the global energy minima (**E3-MD1w** and **E3-MD1o**).

Solvent	Minimum Energy	%B	RMSd (Å)	Intramolecular Interactions ^1^
Water	**E3-MD1w**	27.97	-	π-π
	**E3-MD2w**	26.96	1.71	No
	**E3-MD3w**	10.82	0.65	No
	**E3-MD4w**	10.61	1.82	No
Octanol	**E3-MD1o**	96.87	-	1HB

^1^ HB: hydrogen bond; π-π: stacking interactions.

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
