# Peer review of "In Silico Food-Drug Interaction: A Case Study of Eluxadoline and Fatty Meal"

_ijms, 2020, doi:10.3390/ijms21239127_

Round 1

Reviewer 1 Report

The authors report a nice exploration of the differential adsorption of loperamide and eluxadoline at different pH values of the intestinal tract in the presence of a fatty meal. The problem was addressed with computational techniques, e.g.molecular dynamics, logP predictions, pKa predictions. All relevant forms (conformations/protonated states) in the investigated pH range was thoroughly discussed. A structural explanation about the different absorption of the two drugs is hypothesized based on the computational results. 

Overall the study is well conducted, original and interesting in many aspects, including the highlighted potential of medicinal chemistry techniques in helping deciphering pharmacokinetics data from structural grounds.

I strongly recommend publication after some revisiting of the English.

Author Response

We thank the reviewer for his/her appreciation. We send the manuscript after some revisiting of the English.

Reviewer 2 Report

The work of Maruca et al. is a deep study of the interaction of food-drug interaction.

The work is very innovative and well described.

I think is suitable for publiccation.

Author Response

(The authors gave the same response as above.)
